# Programming nonreciprocity and reversibility in multistable mechanical metamaterials

Gabriele Librandi[1], Eleonora Tubaldi [2 ✉] & Katia Bertoldi [1 ✉]

Nonreciprocity can be passively achieved by harnessing material nonlinearities. In particular, networks of nonlinear bistable elements with asymmetric energy landscapes have recently been shown to support unidirectional transition waves. However, in these systems energy can be transferred only when the elements switch from the higher to the lower energy well, allowing for a one-time signal transmission. Here, we show that in a mechanical metamaterial comprising a 1D array of bistable arches nonreciprocity and reversibility can be independently programmed and are not mutually exclusive. By connecting shallow arches with symmetric energy wells and decreasing energy barriers, we design a reversible mechanical diode that can sustain multiple signal transmissions. Further, by alternating arches with symmetric and asymmetric energy landscapes we realize a nonreciprocal chain that enables propagation of different transition waves in opposite directions.

[1] John A. Paulson School of Engineering and Applied Sciences, Harvard University, Cambridge, MA, USA. [2] Department of Mechanical Engineering, University of Maryland, College Park, MD, USA. ✉email: etubaldi@umd.edu; bertoldi@seas.harvard.edu

N onreciprocity—asymmetric transmission of energy between any two points in space—is receiving increasing interest in many areas of physics[1,2], including optics[3,4], electromagnetism[5,6], elasticity[7,8], and acoustic[9–12]. Focusing on elastic systems, nonreciprocity has been successfully exploited to realize selective signal transmission[13–17], logic elements[18,19], direction-dependent insulators[20,21], and switches[22]. To achieve such remarkable behaviors, both active and passive strategies have been proposed. On the one hand, nonreciprocity for linear waves has been obtained either by imparting a rotation to the medium[23] or by introducing activated materials with time-modulated properties in space and time[24–26] to break time-reversal symmetry. On the other hand, nonreciprocity has also been demonstrated in passive media by harnessing nonlinear phenomena[27–30]. In particular, mechanical metamaterials with two or more stable equilibrium states have recently emerged as a powerful platform to realize nonreciprocity, as they support only unidirectional transition wave propagation when comprising an array of bistable building blocks with asymmetric energy wells[19,31–34]. However, although this strategy is appealing for its simplicity and robustness, it typically leads to nonreversible wave propagation since these systems release a net amount of energy upon propagation of the pulses and needs to be manually "recharged" (i.e., all elements need to be reset to their higher energy well) to sustain a second wave.

Here, we demonstrate the realization of a multistable mechanical metamaterial for which nonreciprocity and reversibility can be independently programmed. Such control of the dynamic response is made possible by the rich and highly tunable behavior of shallow arches, as their energy landscape can be easily adjusted to exhibit target energy barriers as well as symmetric or asymmetric wells. We first show that chains comprising identical arches with symmetric energy wells support the propagation of nonlinear pulses that sequentially switch the elements to their inverted stable configuration. However, although such signal propagation is reciprocal and reversible, it is not stable (Fig. 1a), as the wave evolves during propagation. Then, we demonstrate that by carefully designing the arches and their arrangement to break symmetry either at the structural or element level, we can enable not only stable propagation of the signal but also a wide range of nonreciprocal behaviors. For example, a reversible diode can be created by connecting shallow arches with symmetric but graded on-site energy potentials (Fig. 1b). Further, a tunable 1D nonreciprocal chain, which enables propagation of different transition waves in opposite directions, can be obtained by alternating shallow arches with symmetric and asymmetric energy potentials (Fig. 1c). As such, our work opens avenues for the design of the next generation of nonlinear structures and devices with robust, nonreciprocal elastic wave-steering capabilities.

## Results

**Symmetric elements—symmetric array**. We consider 1D chains comprising $N$ shallow arches connected via rotating hinges that impose continuity of rotations between adjacent elements. All arches have end-to-end distance $L = 120$ mm and are made of spring steel shims with thickness $h = 0.3048$ mm, width $b = 10$ mm, length $l \in [103.1, 105.0]$ mm, volumetric density $\rho = 7850$ kg/m$^3$ and Young's modulus $E = 170$ GPa (see Supplementary Information "Fabrication" section for details). To excite the system, we move with an indenter the midpoint of either the first or last arch in the array at a constant speed $\alpha = 15$ mm/s. We then monitor the response of the chain with a high-speed camera and track the position of the central point of the $j$-th arch, $w_j(L/2, t)$, as a function of time $t$ (see Supplementary Information "Testing" section for details).

We start by focusing on an array comprising three arches (i.e., $N = 3$) with rise $e_j = w_j(L/2, t = 0) = 12.4$ mm ($j = 1$, 2, 3) realized by elastically buckling flat metallic shims of length $l = 105$ mm (see Supplementary Information "Fabrication" section for details). The results reported in Fig. 2a, b for a test in which the indenter acts on the leftmost arch show two key features. First, as recently observed for individually hinged arches under displacement control[35], the indenter makes the leftmost arch snap to its symmetric stable configuration through the activation of the first asymmetric deformation mode (see Fig. 2a). Second, and most important, this reconfiguration does not remain localized as the energy released by the arch upon snapping is transmitted to the neighboring element through the rotation of the hinges. As a result, the snapping of the first arch triggers a cascade of snapping events that sequentially switches the other two elements to their symmetric stable configuration (see Supplementary Movie 1). This response is fully reciprocal and reversible since actuating the first or the last arch (i.e. left-to-right vs. right-to-left), from the top or the bottom (i.e., up-to-down vs. down-to-up) always produces the same dynamic behavior (see Supplementary Fig. 8 for details). However, if a fourth arch is added to the chain (i.e., for $N = 4$), the input provided by the indenter is not sufficient to generate a signal that switches all the elements of the lattice (Fig. 2c, d, see Supplementary Movie 1).

**Predictive numerical model**. To get a deeper understanding of the snapping signal transmission through the chain, we establish a numerical model. We focus on the $j$-th arch, use Euler-Bernoulli beam theory[36] to describe its response[35,37–42] and impose continuity of rotations between neighboring elements[43,44]

$$\frac{\partial w_{j-1}}{\partial x_{j-1}}\bigg|_{x_{j-1}=L} = \frac{\partial w_j}{\partial x_j}\bigg|_{x_j=0}, \tag{1}$$

where $x_j \in [0, L]$ represents the local axial coordinates and $w_j(x_j, t)$ denotes the time-dependent profile of the $j$-th arch. Importantly, the constraints described by Eq. (1) introduce concentrated moments at both ends of the $j$-th arch, $M_{L_j}$, and $M_{R_j}$ (Fig. 2e), and these satisfy

$$M_{R_{j-1}} = -M_{L_j}, \qquad M_{R_j} = -M_{L_{j+1}}. \tag{2}$$

It follows that the response of a chain comprising $N$ arches can be described by

$$\rho A \frac{\partial^2 w_j}{\partial t^2} + \beta \frac{\partial w_j}{\partial t} + EI\left(\frac{\partial^4 w_j}{\partial x_j^4} - \frac{d^4 w_{0j}}{dx_j^4}\right) + p_j \frac{\partial^2 w_j}{\partial x_j^2} +$$
$$Q_1 \delta_{1j} + M_{Lj}\left(1 - \delta_{1j}\right) + M_{Rj}\left(1 - \delta_{Nj}\right) = 0 \tag{3}$$
$$\text{for} \quad j = 1, ..., N$$

where $A$ and $I$ are the area and moment of inertia of the arches' cross-section, $\rho$ and $E$ are the volumetric density and Young's modulus of the material, $\beta$ represents the viscous damping coefficient and $\delta_{1j}$ and $\delta_{Nj}$ are Kronecker delta functions. Moreover, $Q_1$ is the reaction force measured at the indenter, and $w_{0j}$ and $p_j$ are the initial unstressed position of the midsurface and the midplane force produced by the stretching of the midsurface of the $j$-th arch, respectively (see Supplementary Information "Mathematical model" section for more details). For our system, $w_j(x_j, t)$ can be expressed as a series of sine functions[35]

$$w_j(x_j, t) = w_{0j}(x_j) + \sum_{n=1}^{N_t} \psi_{nj}(t) \sin\left(\frac{n\pi x_j}{L}\right). \tag{4}$$

Substitution of Eq. (4) into Eqs. (3) leads to a system of $N_t \times N$ coupled ordinary differential equations that we numerically solve to obtain the modal amplitudes $\psi_{nj}$. As shown in Fig. 2b, d, the numerical predictions are in very good agreement with the

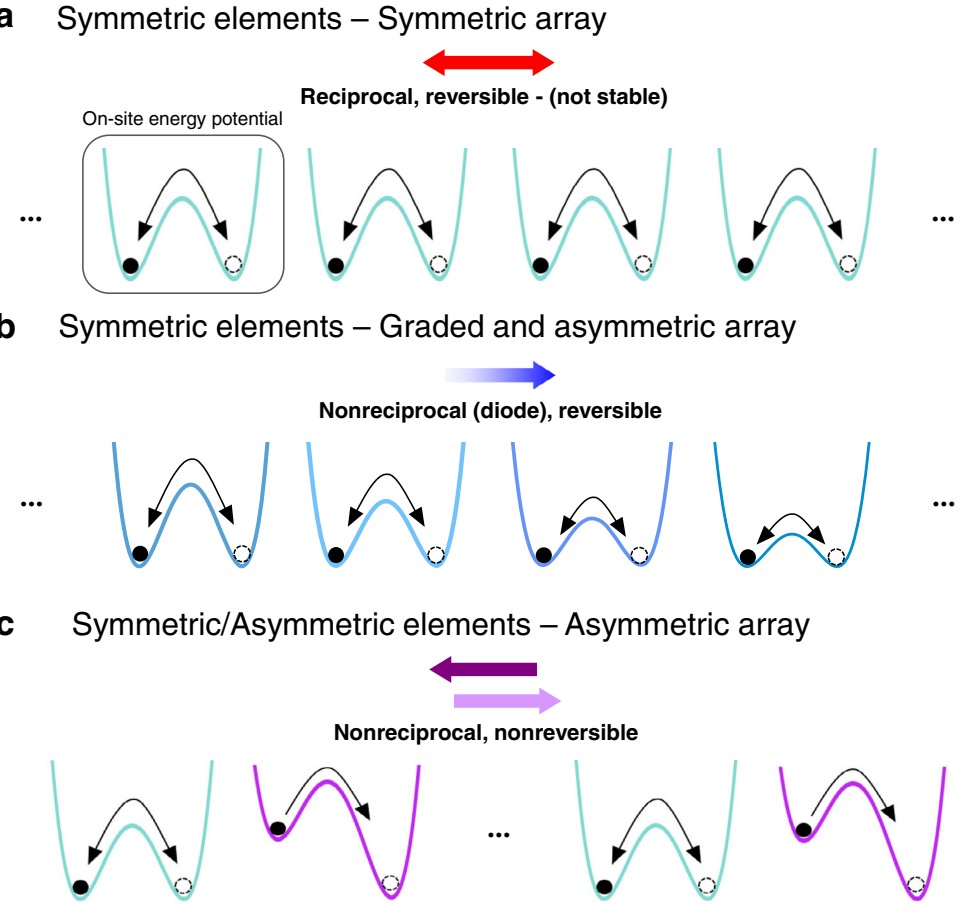

**a** Symmetric elements – Symmetric array

Reciprocal, reversible - (not stable)

On-site energy potential

**b** Symmetric elements – Graded and asymmetric array

Nonreciprocal (diode), reversible

**c** Symmetric/Asymmetric elements – Asymmetric array

Nonreciprocal, nonreversible

**Fig. 1 Programming nonreciprocity and reversibility. a** Signal propagation in a chain comprising identical bistable elements with symmetric energy wells is reciprocal and reversible, but not stable. **b** Signal propagation in a chain comprising bistable elements with symmetric energy wells, but decreasing energy barriers is nonreciprocal and reversible. **c** Signal propagation in a chain comprising bistable elements with both symmetric and asymmetric energy wells is nonreciprocal and nonreversible.

experimental results when choosing $N_t = 3$ and $\beta = 1.4\,\text{kg}/(\text{m·s})$ (see Supplementary Information "Mathematical model" section for details and Supplementary Movie 2) and capture both the propagation of the snapping signal through the entire chain for $N = 3$ and its arrest for $N = 4$.

To understand the absence of a stable propagation in the system comprising $N = 4$ elements, we focus on a single hinged arch and use Euler-Bernoulli beam theory to determine its energy landscape when one of its ends is forced to rotate (see Supplementary Information 'Mathematical model' for details). As observed in our experiments, we find that it is energetically more favorable for the arches to activate the first antisymmetric mode when snapping between the two stable states (see cyan path in Fig. 2f). However, despite the asymmetric deformation path, the on-site energy potential of the arches is symmetric and characterized by two wells of equal height at $w_{L/2} = \pm e_j$ separated by an energy barrier $V^b = 26\,\text{mJ}$. As such, there is no net-release of energy when the arches snap between their two stable configurations, and the stable propagation of the snapping signal is only possible in unrealistic systems without any form of dissipation (see numerical results for $\beta = 0$ in Supplementary Fig. 9).

**Symmetric elements—graded and asymmetric array.** To achieve stable wave propagation as well as to independently control reciprocity and reversibility, we then introduce asymmetry into

the system both at the structural and arch levels. To begin with, we build a 1D non-symmetric array by assembling elastically deformed shallow arches with monotonically decreasing rises (Fig. 3a and Supplementary Table 1 for details). Since the energy barrier $V^b$ monotonically decreases as $e_j$ becomes smaller (Fig. 3b, c), the effect of dissipation can be counteracted by tuning the rises to make $\Delta V_j^b = V_{j-1}^b - V_j^b$ larger than the energy dissipated by the $j$-th arch during snapping. As shown in Fig. 3d, when the indenter excites the leftmost arch with the highest rise, a stable snapping wave propagates from left-to-right. Importantly, such wave propagation is reversible and nonreciprocal. Since all the arches have a symmetric energy landscape, the wave can be excited by snapping the first arch both up-to-down and down-to-up and there is no need to "manually" recharge the system to propagate a new signal (i.e., the behavior is fully reversible—see Supplementary Fig. 10). However, when the indenter excites the arch with the lowest rise the energy released upon its snapping is not enough to make the next arch jump. As such, there is no wave propagation from right-to-left and the system acts as a mechanical diode (see Supplementary Movie 3).

Although in Fig. 3d, we focus on a specific system with $N = 10$ and $\Delta e_j = e_j - e_{j+1} \sim 500\,\mu\text{m}$, we next use our model (which nicely captures the experimental results of Fig. 3d) to systematically investigate the effect of $\Delta e_j$ on the signal propagation in 1D arrays of graded shallow arches. We find that $\Delta e_j$ plays a very important role, as it directly affects the difference in an energy

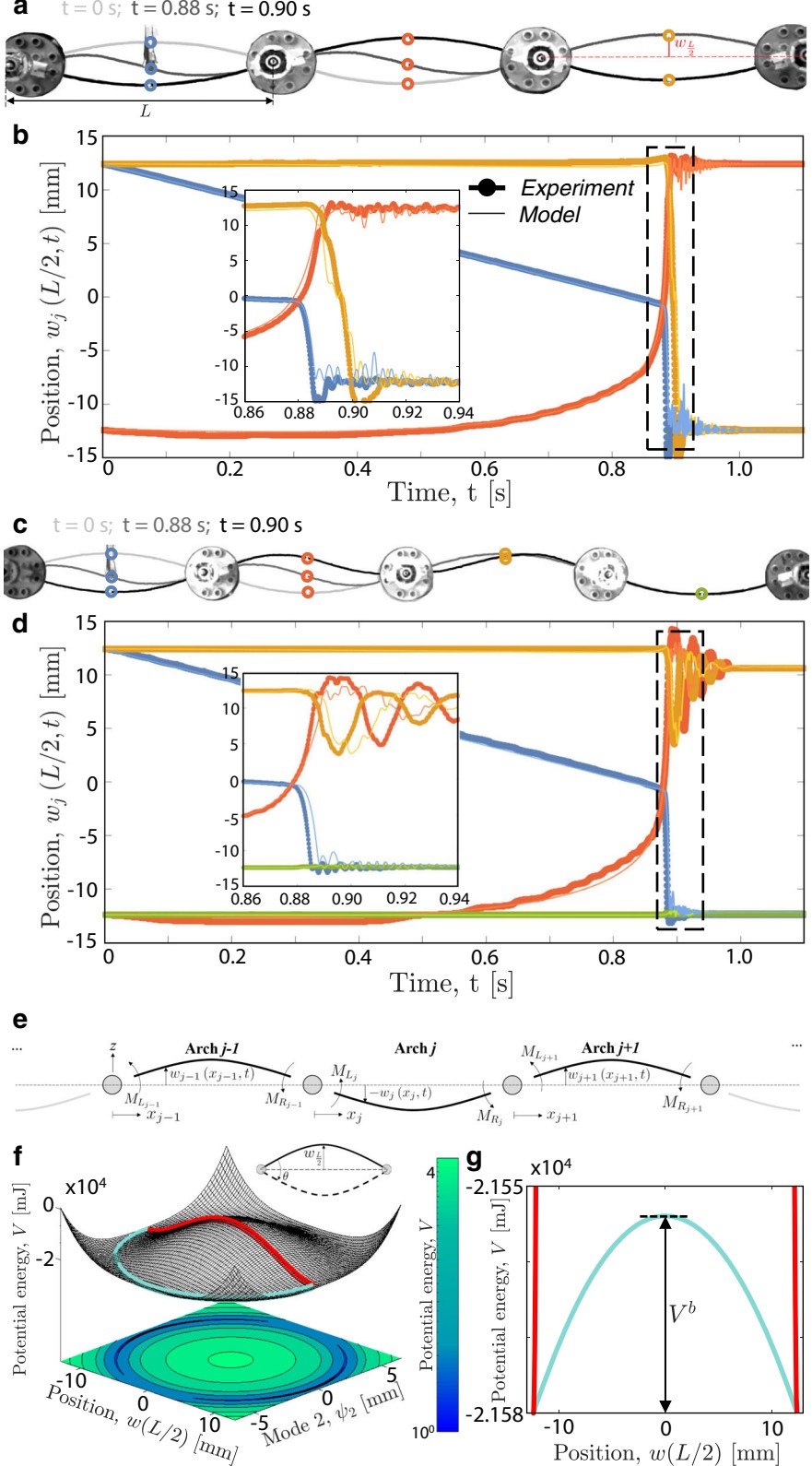

**Fig. 2 Symmetric elements—symmetric array. a**, **b** Array comprising three identical arches with rise $e_j = 12.4$ mm ($j = 1, 2, 3$) and symmetric energy wells. **a** Snapshots at $t = 0$ s, 0.88 s, 0.90 s and **b** evolution of the positions of the midpoints of the arches, $w_j(L/2, t)$, for a test in which the indenter acts on the leftmost arch. Thick-dotted and thin lines correspond to experimental and numerical results, respectively. **c**, **d** Same as **a**, **b** but for an array comprising four arches. **e** Schematic of the system. **f** On-site energy potential for an elastically deformed shallow arch as a function of $\psi_2$ and $w(L/2)$. Note that $w(L/2) = \psi_1 - \psi_3$ (see Eq. 4). The red line indicates a deformation path along which only the first symmetric mode is activated. The cyan line corresponds to the minimum energy path. **g** Comparison between the red and cyan paths shown in **f**, highlighting the symmetry of the two energy minima and the energy barrier, $V^b$, separating them.

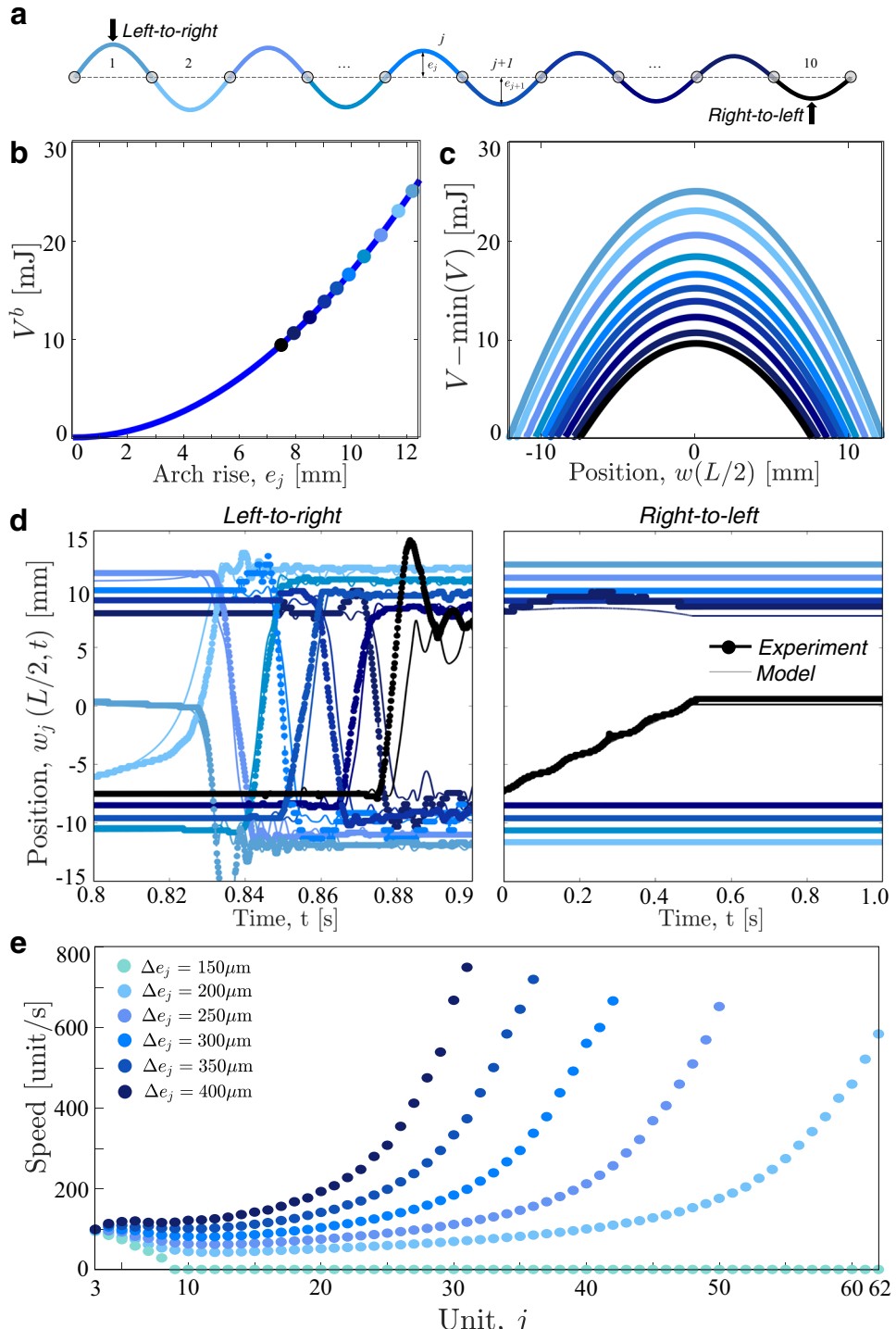

**Fig. 3 Symmetric elements—graded and asymmetric array. a** Schematic of the system with $N = 10$ arches with decreasing rises. **b** Energy barrier, $V_b$, versus arch rise, $e_j$. **c** Normalized on-site energy potential, $V - \min(V)$, versus the position of the arch midpoint. **d** Evolution of the positions of the midpoints of the arches, $w_j(L/2, t)$, for tests in which the indenter acts on the leftmost (left-to-right) and rightmost (right-to-left) arches (see Supplementary Fig. 19 for corresponding numerical contour maps). Thick-dotted and thin lines correspond to experimental and numerical results, respectively. **e** Local speed of the transition wave along the chain for different values of $\Delta e_j$ as predicted by our model.

barrier between neighboring elements, $\Delta V_j^b$ (see Fig. 3b). More specifically, the numerical results reported in Fig. 3e show that for $\Delta e_j \leq 150\,\mu m$, the velocity of the transition wave (calculated by monitoring the time at which the arches reach the inverted stable configuration—see Supplementary Fig. 11 for details) monotonically decreases during propagation and eventually vanishes.

For such small values of $\Delta e_j$, $\Delta V_j^b$ is not sufficient to overcome the effects of dissipation of the system, and stable wave propagation is not supported. By contrast, for $\Delta e_j > 150\,\mu m$ the difference in energy barriers between consecutive arches is larger than the dissipation upon snapping and the signal propagates through the entire chain. Further, our numerical results indicate that the

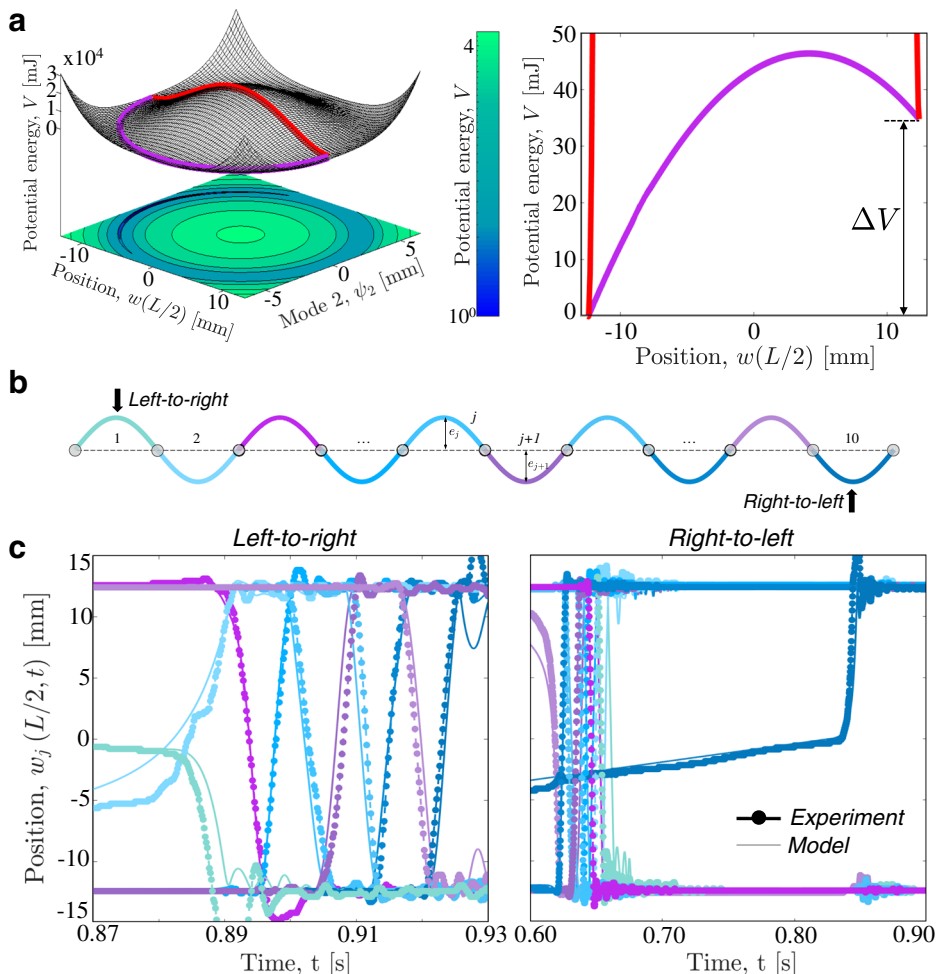

**Fig. 4 Symmetric/asymmetric elements—asymmetric array. a** On-site energy potential for a plastically deformed shallow arch. The red line indicates a deformation path along which only the first symmetric mode is activated. The purple line corresponds to the minimum energy path. **b** Schematic of the system with $N = 10$ shallow arches and one plastically deformed arch arranged every two elastically deformed ones. **c** Evolution of the positions of the midpoints of the arches, $w_j(L/2, t)$, for tests in which the indenter acts on the leftmost (left-to-right) and rightmost (right-to-left) arches (see Supplementary Fig. 19 for corresponding numerical contour maps). Thick-dotted and thin lines correspond to experimental and numerical results, respectively.

waves accelerate during propagation. This is because the energy that the arches need to absorb to overcome the energy barrier and snap monotonically decreases along the chain, causing a faster transition rate.

**Symmetric/asymmetric elements—asymmetric array.** The results of Fig. 3 indicate that graded 1D arrays of shallow arches with symmetric energy landscape can support stable, reversible, and unidirectional propagation of transition waves. Next, to achieve additional control on nonreciprocity, we introduce elements with asymmetric on-site energy potential. Such asymmetry at the arch level can be easily realized by plastically deforming the metallic shim into the target shape, $e_j \sin(\pi x/L)$. As shown in Fig 4a, the plastic deformation makes the two energy minima different. Specifically for arches with $e_j = 12.4$ mm, our model predicts that the transition between the two stable states involves a net change $\Delta V = 34.8$ mJ in stored potential energy. Depending on the direction of the transition, the arch either absorbs energy[45] or releases it, enabling unidirectional propagation of transition waves over long distance[19,32]. Although previous studies have considered arrays of purely asymmetric elements[19,31–33], here we investigate the dynamic response of systems comprising a mixture of symmetric and asymmetric elements. This is possible

because the plastically and elastically deformed arches, despite their different energy landscape, share the same shape and, therefore, are geometrically compatible and can be easily combined to form arrays. For example, in Fig. 4b we consider a chain comprising seven arches with symmetric on-site potential (i.e., elastically deformed arches—see blue arches in Fig. 4b) and three with asymmetric energy profile (i.e., plastically deformed arches—see purple arches in Fig. 4b) set in their higher energy well—all with rise $e_j = 12.4$ mm. When the arches are arranged as in Fig. 4b (with one plastically deformed arch every two elastically deformed ones), the energy released upon snapping by the plastically deformed arches enables signal propagation through the entire array both left-to-right and right-to-left (see Fig. 4c). However, because of the structural asymmetry of the chain, the energy is released by the asymmetric elements at different locations when the wave travels left-to-right and right-to-left, leading to different signal propagation in the two directions. When the indenter acts on the leftmost unit, the second arch reaches the inverted stable configuration at $t_2^{\text{snap}} = 0.89$ s, whereas the last one snaps at $t_{10}^{\text{snap}} = 0.93$ s. By contrast, when the rightmost unit is excited, the pulse is initiated at $t_9^{\text{snap}} = 0.62$ s and arrives at the other end of the chain at $t_1^{\text{snap}} = 0.66$ s. Interestingly, while for left-to-right propagation the arches snap in sequence (i.e., the

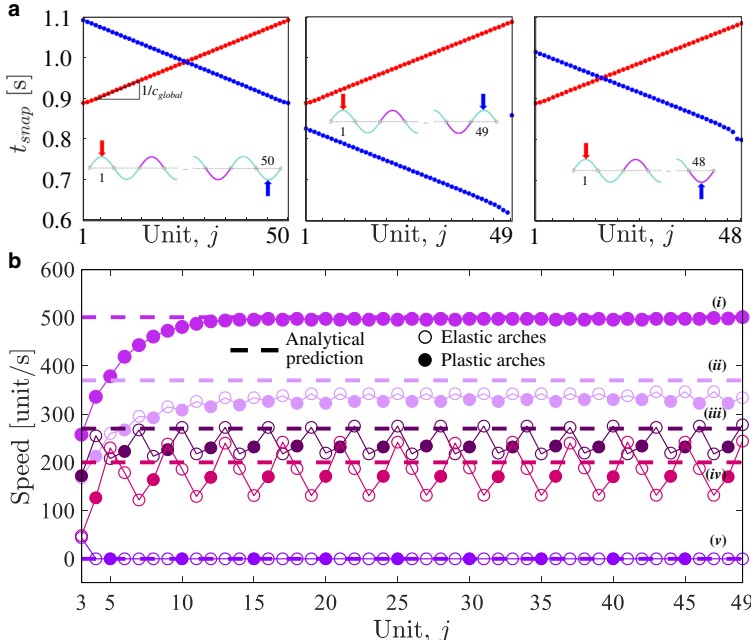

**Fig. 5 Tuning nonreciprocity and wave speed. a** Snapping times for transition waves propagating both left-to-right and right-to-left in chains comprising $N$ = 48, 49, and 50 elastically and plastically deformed arches periodically arranged according to the pattern shown in Fig. 4b. Note that for $N$ = 49 the arches do not snap in sequence when excited right-to-left (the blue dot representing the snapping time for the 49-th arch is $t_{49}^{snap}$ = 0.86 s. **b** Local speed of the transition waves along the chain for different patterns of elastically/plastically deformed arches. The dashed lines correspond to the predictions of Eqs. (5–7). (i) Chain with all plastic arches, (ii) chain with one plastic and one elastic arches, (iii) chain with one plastic and two elastic arches, (iv) chain with one plastic and three elastic arches, (v) chain with one plastic and four elastic arches.

leftmost arch snaps first and the rightmost one snaps as last), for right-to-left propagation the arch excited by the indenter is the last one to snap at $t_{10}^{snap}$ = 0.85 s. Finally, it is important to note that the signal propagation in this system is nonreversible as the plastically deformed arches can only snap from the high energy well to the lower energy well. As such, the chain needs to be manually recharged to support the propagation of a new signal (see Supplementary Fig. 13 and Supplementary Movie 4).

Next, to better understand how the global structural asymmetry affects the nonreciprocity of wave propagation, we numerically investigate the response of chains comprising $N$ = 48, 49, and 50 (Fig. 5a and Supplementary Fig. 14) elastically and plastically deformed arches periodically arranged according to the pattern shown in Fig. 4b. We find that for all considered chains the pulses propagate at a speed $c_{global}$ ~ 243 unit/s in both directions. However, the time at which the signal is initiated for left-to-right and right-to-left propagation can be programmed by altering the asymmetry of the chain through $N$. More specifically, in a symmetric chain with $N$ = 50 the snapping signal is initiated at the same time for both propagation directions (i.e., $t_2^{snap}$ = 0.89 s and $t_{49}^{snap}$ = 0.89 s for left-to-right and right-to-left propagation, respectively). Differently, for $N$ = 49 and 48 the system is asymmetric (as there are either one or no elastically deformed arches separating the rightmost plastically deformed one from the right end) and the wave starts at $t_{48}^{snap}$ = 0.62 s and $t_{47}^{snap}$ = 0.80 s when the rightmost arch is excited.

While asymmetry enables as to tune the time at which the pulses are initiated from the left and right end, control on the speed of the pulses can be achieved by varying the density of plastically deformed elements in the chain. To demonstrate this point, in Fig. 5b we report the numerically predicted velocity for left-to-right propagation in chains with $N$ = 49 plastically and elastically deformed arches arranged according to different periodic patterns. First, we find that stable wave propagation is

only possible when the plastically deformed arches are separated by three or less elastically deformed ones. Second, the results indicate that $c_{global}$ monotonically increases with the density of plastically deformed arches and approaches ~497 units/s in the limit of a chain comprising only plastically deformed elements. Note that $c_{global}$ can be evaluated by balancing the total transported kinetic energy, $E_d$, the difference $\Delta V$ between the higher and lower energy well for the asymmetric elements, and the energy dissipated as ref. [46]

$$c_{global} = \frac{2\beta L E_d}{\Delta V}. \tag{5}$$

where $E_d$ can be estimated as

$$E_d = \sum_{j=1}^{n_{sc}} \frac{1}{2} \left( \frac{1}{n} \sum_{i=1}^{n} v_{(j-1)n+i} \right)^2 nL, \tag{6}$$

$n$ denoting the number of arches in the super-cell that captures the periodic pattern of elastic/plastic arches and $n_{sc}$ being the number of super-cells in the chain. Moreover, $v_j$ is the speed at which the $j$-th arch snaps (i.e., the snapping-speed), which can be computed as

$$v_j = \frac{|w_j(L/2, t_{in})| + |w_j(L/2, t_{end})|}{t_{end} - t_{in}} \tag{7}$$

$t_{in}$ and $t_{end}$ denoting the instants of time at which the snapping of the $j$-th arch starts and ends, respectively (see Supplementary Fig. 15). In Fig. 5b we compare the predictions from Eq. (5) with the numerical results for chains with varying density of plastically deformed elements and find that the model nicely captures the wave speed observed in the simulations. Finally, it is interesting to point out that the alternation of elastically and plastically deformed elements leads to pulses with locally modulated speed. This is because the energy released upon snapping by the plastically deformed arches makes the following elastic element to

snap faster, whereas the absence of released energy between consecutive elastically deformed ones delays their snapping.

## Discussion

To summarize, we have shown that in 1D multistable systems nonreciprocity and reversibility can be programmed independently and easily realized. A reversible diode can be created by assembling elements with symmetric on-site energy potentials but decreasing energy barriers. On the other hand, chains capable of sustaining nonreciprocal transition waves traveling in opposite directions can be realized by alternating arches with symmetric and asymmetric energy wells. Although the dynamic response of the reversible diode is controlled by the difference in rise between the arches, the behavior of the nonreciprocal chain can be tuned by varying the arrangement of the symmetric and asymmetric elements and is negligibly affected by their rise (see Supplementary Fig. 16). Further, all the considered systems are input-independent, as the speed of the supported waves is insensitive to the loading rate $\alpha$ at which the indenter moves the first arch (see Supplementary Fig. 17). Although in this study we verified the concept for 1D chains, our findings can be easily generalized to 2D and 3D networks of arches to realize passive smart systems with enhanced selective signal transmission and wave guidance capabilities.

## Methods

Details on the geometry, design, fabrication, testing, analytical model, and numerical solutions of the 1D chains comprising shallow arches are provided in the Supplementary Information.

## Data availability

The experimental and numerical data in support of the findings in this study are available from the corresponding author upon request.

## Code availability

All numerical codes used to study the nonreciprocity and reversibility of 1D chains of shallow arches are available from the corresponding author upon request.

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

## Acknowledgements

K.B. acknowledges support from NSF grants EFMA-1741685 and DMR-2011754 and Army Research Office Grant W911NF-17-1-0147. E.T. acknowledges support from the University of Maryland, College Park - startup package. G.L. and E.T. acknowledge John Hutchinson and Lakshminarayanan Mahadevan for interesting conversations and helpful suggestions.

## Author contributions

G.L., E.T, and K.B. designed research and analyzed data; G.L. performed experiments; G.L. and E.T. developed the model; E.T. and K.B. jointly supervised this work; G.L., E.T, and K.B. wrote the paper.

## Competing interests

The authors declare no competing interests.
