## [Peer Review File · Nature Communications]

REVIEWER COMMENTS

Reviewer #1 (Remarks to the Author):

This manuscript presents the study of the propagation of pulses in non-dimensional arrays of bistable arches using both experiments and a numerical model. The influence of asymmetry of the energy potential, and of introducing gradients in the energy of individual arches on non-reciprocity and reversibility is analyzed. This is an interesting study with some novel results that provides new insights into the nonlinear dynamics of chains of bistable elements. While this is an interesting paper, the presented research will have a bigger impact if some additional explanations that go beyond a description of the experimental or numerical results were added (particularly related the last 2 figures). Another sources of improvement to the manuscript would be to improve the quality of some of the figures

1. To make the manuscript more accessible to a broad audience, it would be useful to clearly define “stable wave propagation” the first time it is used, in the last paragraph of page 1, precisely.
2. I think that the terminology “asymmetric array” is imprecise (Fig. 1b and Fig. 3). I believe that what is needed is more restrictive than just asymmetry; a gradient in the height of the barrier is required to obtain a “diode”. The figure and results are all based on using a monotonic decrease in the height of the energy barrier.
3. If EI is used in Eq. 3, this is not Kirchhoff thin plate theory but Euler-Bernoulli beam theory.
4. Some clarifications are needed for the potential energy figure (Fig. 2f). The energy is shown as a function of two variables. However, if there are 3 modes in the solutions ($Nt=3$), the energy is a function of three variables. Was an additional constraint applied for this figure?
5. Some figures (Fig. 4, and in supplemental information S10, S13, S13) use a single color for multiple lines, making it impossible to identify which line corresponds to a given arch. I suggest perhaps a gradient of color, such that the reader can more easily read the graph and identify the sequence.
6. I find that the section of the results with the chain with symmetric and asymmetric potentials is only descriptive, and lacks good physical explanations about why the system behaves as described. Why does going from 50 to 49 cause the chain to snap at a much earlier time in the right-to-left direction? Why does a chain with only asymmetric elements have a faster propagation speed than a chain with both asymmetric and symmetric elements?
7. Page 7 the authors mention that for right-to-left propagation the arch excited by the indenter is the last one to snap for the results of Fig. 4. However, in the results of Fig. 5a with longer chains, it seems that the arches always snap in sequence. Why is there a different behavior?
8. The nonreciprocity obtained by introducing asymmetry in chains by going from 50 to 49 or 48 arches is very similar (but not identical) to what Wu and Wang have shown in a recent paper with different types of chains with bistable elements. (Wu, Z., Zheng, Y. and Wang, K.W., 2018. Metastable modular metastructures for on-demand reconfiguration of band structures and nonreciprocal wave propagation. *Physical Review E*, 97(2), p.022209.)
9. What is the last term in Eq. S2 divided by $8L$, and not $2L$?
10. I believe that in Supplemental Information, after Eq. S8: “until” should be “while”

11. Fig. S9 is difficult to understand. I am not sure what I am looking at
12. Fig. S10: which lines are on the left/right side of the chain? Are the lines offset?

Reviewer #2 (Remarks to the Author):

The paper "Programming nonreciprocity and reversibility in multistable mechanical Metamaterials" by Librandi et al. reports the non-reciprocal and reversible transmission of nonlinear mechanical waves in mechanical systems consisting of bistable arches with spatially graded (a)symmetric energy barriers. In particular, the authors demonstrate that such mechanical systems can transmit nonlinear waves unidirectionally and reversibly using spatial grading. The authors further investigate the dynamics of these systems, they notably demonstrate that the nonlinear waves tend to accelerate. They also explore the design space by investigating how asymmetric bistable potentials can be used as energy sources to overcome dissipation. The paper is overall nicely written and well structured and these results will be of interests to scientists interested in nonlinear waves in mechanical systems. However, although the specific platform used by the authors is new, they are not the first to investigate spatially graded structures or asymmetric potentials and unfortunately, I don't see where the conceptual novelty is. For these reasons, I don't think that the paper is a good fit for Nat. Comm. and I would recommend a more specialized publication venue. Please see my comments below.

1. Transition waves in spatially graded bistable architectures with non-reciprocal wave propagation have been discussed in earlier works, see e.g. Hwang and Arrieta, PRE 2018. As acknowledge by the authors, transition waves with asymmetric potentials have been already discussed in refs [19-29,31]. In comparison to these works, I unfortunately don't see much conceptual novelty in the present work.
2. I think that a homogenized description that would describe the continuum limit governing equation for $\psi_1(j,t)$ could be useful to predict the dynamics of transition wave and rationalize the various findings discussed in the paper. I feel that a description at such level is now lacking.
3. I find the figures like Figs. 3d, 4c hard to parse. Contour plots would perhaps be easier to read? Also, I find Fig. 5b hard to parse with symbols having same colors in different curves.

Reviewer #3 (Remarks to the Author):

Reciprocity is a generally accepted concept in classic elastic systems. However, recent studies on bistable and multistable mechanical metamaterials have shown that nonreciprocity can be realized by careful design of the microstructure constituting the metamaterials. This study demonstrates via 1D arrays of bistable arches that nonreciprocity and reversibility in the elastic wave propagation can be independently tuned. The idea is to combine the symmetry in the energy potential at element level and

the geometry symmetry at structural level. Although the stable propagation of signals is a direct extension of the authors' PNAS paper (Ref. 19), the demonstrated reversible and nonreversible nonreciprocal behaviors via elastically and plastically deformed arches are interesting. The paper is thus recommended for publication in Nature Communications.

The following comments can be considered by the authors.

(1) In Figs. 2-4, the authors showed excellent agreement between experiment and modeling, with a particular choice of the viscous damping factor being 1.4×10^{-6} kg/(ms). It is not clear how the authors adopted such a value. Is it simply by the trial and error method?

(2) It was shown in Fig. 3e that stable propagation of signals is obtainable when the rise $\Delta \langle e_j \rangle$ is great than $150 \mu\text{m}$. The underlying mechanism is that stable propagation is possible when the dissipated energy of each unit can be fully compensated by the energy difference between adjacent units. It would be interesting if the authors can quantify relationship between the critical value of $\Delta \langle e_j \rangle$ and the viscous damping factor.

(3) The authors claimed that "control on the speed of the pulses can be achieved by varying the density of plastically deformed elements in the chain". This is true by considering the potential energy shown in Fig. 4a for a plastically deformed arch. The energy difference $\Delta \langle V \rangle$ provides the driving source for the stable propagation. If the authors correlate $\Delta \langle V \rangle$ with the dissipated energy in the system, they should be able to derive an analytical model for the speed of pulses.

Response to the Reviewers' comments for manuscript
(Tracking #: NCOMMS-20-38142):
“Programming nonreciprocity and reversibility in multistable
mechanical metamaterials”

In the following, we address each of the Reviewers' comments/suggestions (*in italic*). In addition, while using the original numbering of figures/equations/references in the manuscript (*e.g.*, Fig. 1, Fig. 2, ..., (1), Eq. (2), ..., [1], [2], ...), we add prefix “*R*” for figures/equations/references presented in this response to the Reviewers' comments (*e.g.*, Fig. R1, Fig. R2, ... Eq. (R1), Eq. (R2), ..., Ref. [R1], Ref. [R2],...).

Response to Reviewer 1

This manuscript presents the study of the propagation of pulses in non-dimensional arrays of bistable arches using both experiments and a numerical model. The influence of asymmetry of the energy potential, and of introducing gradients in the energy of individual arches on non-reciprocity and reversibility is analyzed. This is an interesting study with some novel results that provides new insights into the nonlinear dynamics of chains of bistable elements. While this is an interesting paper, the presented research will have a bigger impact if some additional explanations that go beyond a description of the experimental or numerical results were added (particularly related the last 2 figures).

We thank the Reviewer for his/her insightful and valuable comments, which have enable us to further improve the manuscript.

Comment 1.1

To make the manuscript more accessible to a broad audience, it would be useful to clearly define “stable wave propagation” the first time it is used, in the last paragraph of page 1, precisely.

We thank the Reviewer for pointing this out. Following his/her suggestion we have modified the text in the last paragraph of page 1 to read as

“However, while such signal propagation is reciprocal and reversible, it is not stable (Fig. 1a), *as the wave evolves during propagation.*”

Comment 1.2

I think that the terminology “asymmetric array” is imprecise (Fig. 1b and Fig. 3). I believe that what is needed is more restrictive than just asymmetry; a gradient in the height of then barrier is required to obtain a “diode”. The figure and results are all based on using a monotonic decrease in the height of the energy barrier rises.

We agree with the Reviewer that “asymmetric array” does not fully describe the configuration of the mechanical diode. Following the Reviewer’s suggestion, we have decided to rename it as “graded and asymmetric array”. The main manuscript has been modified accordingly and also the wording in the figures has been modified.

Comment 1.3

If EI is used in Eq. 3, this is not Kirchoff thin plate theory but Euler Bernoulli beam theory

We thank the Reviewer for pointing this out. The Kirchoff-Love theory is an extension of Euler-Bernoulli theory and they share some analogies. However, the Reviewer is right that, since we are using EI , our model falls within the Euler-Bernoulli beam theory. The manuscript has been modified accordingly.

Comment 1.4

Some clarifications are needed for the potential energy figure (Fig. 2f). The energy is shown as a function of two variables. However, it there 3 modes in the solutions ($N_t = 3$), the energy is a function of three variables. Was an additional constraint applied for this figure?

We would like to thank the Reviewer for pointing this out and we do believe this is a valuable point. In Fig. 2f of the main text we indeed report the energy as a function of two variables: the amplitude of Mode 2, ψ_2 , and the position of the midpoint of our shallow arch, $w(L/2)$. However, it follows from Eq. (4) of the main text that $w(L/2)$ depends on the amplitude of both the first and the third mode as

$$w(L/2) = \psi_1 - \psi_3. \quad (\text{R1})$$

Further, since we are applying a rotation on one hinge, the symmetry of the structure is immediately broken and the third mode is practically negligible. For the sake of clarity, we have have modified the caption of Fig. 2 to read as

“Symmetric elements - Symmetric array. a-b, Array comprising three identical arches with rise $e_j = 12.4$ mm ($j = 1,2,3$) and symmetric energy wells. **a** Snapshots at $t = 0$ s, 0.88s, 0.90s and **b**, evolution of the positions of the midpoints of the arches, $w_j(L/2, t)$, for a test in which the indenter acts on the leftmost arch. Thick-dotted and thin lines correspond to experimental and numerical results, respectively. **c-d**, Same as a-b but for an array comprising four arches. **e**, Schematic of the system. **f**, On-site energy potential for an elastically deformed shallow arch as a function of ψ_2 and $w(L/2)$. Note that $w(L/2) = \psi_1 - \psi_3$ (see Eq. (4)). The red line indicates a deformation path along which only the first symmetric mode is activated. The cyan line corresponds to the minimum energy path. **g**, Comparison between the red and cyan paths shown in f, highlighting the symmetry of the two energy minima and the energy barrier, V^b , separating them.”

Comment 1.5

Some figures (Fig. 4, and in supplemental information S10, S13, S13) use a single color for multiple lines, making it impossible to identify which line correspond to a given arch. I suggest perhaps a gradient of color, such that the reader can more easily read the graph and identify the sequence.

As suggested by the Reviewer we have used a gradient of color for different arches in Fig. 4 and Fig. S13 (which are reported below as Figs. R1 and R2 for completeness). We hope that this makes it is easier for the reader to identify the line corresponding to a given arch. Regarding Fig. S10 a gradient of colors was already used in the original version and this gradient was meant to provide indication on the rises, e_j , of different arches. Fig. S14 mainly shows how a signal left-to-right and right-to-left can be initiated at very different instants, therefore we do not believe that being able to track each of the 50 arches would add substantial insights for the reader. Further, on the technical side, finding a palette of color able to unequivocally identify 50 different arches is unpractical.

Figure R1: **Symmetric/Asymmetric elements - Asymmetric array.** **a**, On-site energy potential for a plastically deformed shallow arch. The red line indicates a deformation path along which only the first symmetric mode is activated. The purple line corresponds to the minimum energy path. **b**, Schematic of the system with $N = 10$ shallow arches and one plastically deformed arch arranged every two elastically deformed ones. **c**, Evolution of the positions of the midpoints of the arches, $w_j(L/2, t)$, for tests in which the indenter acts on the leftmost (left-to-right) and rightmost (right-to-left) arches (see Fig. S19 for corresponding numerical contour maps). Thick-dotted and thin lines correspond to experimental and numerical results, respectively.

Symmetric/Asymmetric elements – Asymmetric array

$$|e_j| = |e_{j+1}|$$

Figure R2: **Symmetric/Asymmetric elements – Asymmetric array.** Results for the same structure considered in Fig. 4 of the main text. **a**, Schematics of chain. **b**, Comparison between the experimentally measured (thick-dotted lines) and numerically predicted (thin lines) positions of the midpoints of the arches when the system is excited left-to-right and right-to-left as indicated in (a). **c**, Schematics of the chain. **d**, Comparison between the experimentally measured (thick-dotted lines) and numerically predicted (thin lines) positions of the midpoints of the arches when the system is excited left-to-right and right-to-left as indicated in (c). No transition waves are supported as indicated in (d).

Comment 1.6

I find that the section of the results with the chain with symmetric and asymmetric potentials is only descriptive, and lack good physical explanations about why the system behaves as described. Why does going from 50 to 49 cause the chain to snap at a much earlier time in the right to left direction? Why does a chain with only asymmetric elements have a faster propagation speed than a chain with both asymmetric and symmetric elements?

We thank the Reviewer for rising this point. Following his/her suggestion we have rewritten the text around Fig. 4 that now reads as

“Next, to achieve additional control on nonreciprocity we introduce elements with asymmetric on-site energy potential. Such asymmetry at the arch level can be easily realized by plastically deforming the metallic shim into the target shape, $e_j \sin(\pi x/L)$. As shown in Fig. 4a, the plastic deformation makes the two energy minima different. Specifically, for arches with $e_j = 12.4$ mm our model predicts that the transition between the two stable states involves a net change $\Delta V = 34.8$ mJ in stored potential energy. Depending on the direction of the transition, the arch either absorbs energy [1] or releases it, enabling unidirectional propagation of transition waves over long distance [2, 3]. While previous studies have considered arrays of purely asymmetric elements [2, 3, 4, 5], here we investigate the dynamic response of systems comprising a mixture of symmetric and asymmetric elements. This is possible because the plastically and elastically deformed arches, despite their different energy landscape, share the same shape and, therefore, are geometrically compatible and can be easily combined to form arrays. For example, in Fig. 4b we consider a chain comprising seven arches with symmetric on-site potential (*i.e.* elastically deformed arches - see blue arches in Fig. 4b) and three with asymmetric energy profile (*i.e.* plastically deformed arches - see purple arches in Fig. 4b) set in their higher energy well - all with rise $e_j = 12.4$ mm. When the arches are arranged as in Fig. 4b (with one plastically deformed arch every two elastically deformed ones), the energy released upon snapping by the plastically deformed arches enables signal propagation through the entire array both left-to-right and right-to-left (see Fig. 4c). However, because of the structural asymmetry of the chain, the energy is released by the asymmetric elements at different locations when the wave travels left-to-right and right-to-left, leading to different signal propagation in the two directions. When the indenter acts on the leftmost unit, the second arch reaches the inverted stable configuration at $t_2^{snap} = 0.89$ s, while the last one snaps at $t_{10}^{snap} = 0.93$ s. By contrast, when the rightmost unit is excited, the pulse is initiated at $t_9^{snap} = 0.62$ s and arrives at the other end of the chain at $t_1^{snap} = 0.66$ s. Interestingly, while for left-to-right propagation the arches snap in sequence (*i.e.* the leftmost arch snaps first and the rightmost one snaps as last), for right-to-left propagation the arch excited by the indenter is the last one to snap at $t_{10}^{snap} = 0.85$ s. Finally, it is important to note that the signal propagation in this system is non-reversible as the plastically deformed arches can only snap from the high energy well to the lower energy well. As such, the chain needs to be manually recharged to support propagation of a new signal (see Fig. S13 and Movie S4). ”

We believe that the revised text better explains why the system behaves as described. Further, we want to point out that, by releasing a net amount of energy upon snapping from the higher to the lower energy well, plastically deformed arches allow to both overcome dissipation effects and to increase the propagation speed. Thus, a chain with only asymmetric elements that constantly release stored potential energy to the system has a faster propagation speed than a chain comprising a combination of both symmetric, which do not release any stored energy, and asymmetric elements.

Comment 1.7

Page 7 the authors mention that for right-to-left propagation the arch excited by the indenter is the last one to snap for the results of Fig. 4. However, in the results of Fig. 5a with longer chains, it seems that the arches always snap in sequence. Why is there a different behavior

We thank the Reviewer for addressing this point and we apologize for not having expressed this concept clearly enough. Left-to-right and right-to-left propagations can be made different in chains comprising periodically arranged elastically and plastically deformed arches by introducing asymmetry at the chain level. For example, if we consider a chain with $N = 49$ arches periodically arranged according to the pattern shown in Fig. 4b of the main text, the third unit is the first asymmetric element that the pulse encounters when propagating left-to-right. Differently, for a pulse that propagates right-to-left the first asymmetric element to snap is the second one in the chain. Therefore, since the energy is released by the asymmetric elements at different locations along the chain when the wave travels left-to-right and right-to-left, different signal propagation in the two directions is achieved. By contrast, for $N = 50$ the chain is symmetric at the chain level (*i.e.* the third unit is the first asymmetric element that the pulse encounters when propagating both left-to-right and right-to-left) and left-to-right and right-to-left waves are identical.

Comment 1.8

The nonreciprocity obtained by introducing asymmetry in chains by going from 50 to 49 or 48 arches is very similar (but not identical) to what Wu and Wang have shown in a recent paper with different type of chains with bistable elements. (Wu, Z., Zheng, Y. and Wang, K.W., 2018. Metastable modular metastructures for on-demand reconfiguration of bandstructures and nonreciprocal wave propagation. Physical Review E, 97(2), p.022209.).

We thank the Reviewer for letting us know about this interesting paper, which is extremely relevant for our study. As such, in the revised version of the manuscript it has been cited.

Comment 1.9

What is the last term in Eq. S2 divided by $8L$, and not $2L$?

Eq. (S2) provides the potential energy of the individual arches

$$V_j = \frac{1}{2}EI \int_0^L \left(\frac{\partial^2 w_j}{\partial x_j^2} - \frac{d^2 w_{0j}}{dx_j^2} \right)^2 dx_j - \frac{1}{2}P_j \int_0^L \left(\frac{\partial w_j}{\partial x_j} \right)^2 dx_j + \frac{EA}{8L} \left[\int_0^L \left(\frac{\partial w_j}{\partial x_j} \right)^2 - \left(\frac{dw_{0j}}{dx_j} \right)^2 dx_j \right]^2 \quad (\text{R2})$$

where w_{0j} is the initial unstressed position of the midsurface of the j -th arch, P_j is the axial force applied to the j -th arch to elastically buckle it. Moreover, A and I are the area and moment of inertia of the cross section, E is the Young's modulus of the material.

Note that the last term in Eq. (R2) represents the potential energy due to the midplane stretching, which is given by

$$V_s = \frac{1}{2}S\Delta \quad (\text{R3})$$

where Δ is the total midplace stretching

$$\Delta = \frac{1}{2} \int_0^L \left[\left(\frac{\partial w_j}{\partial x_j} \right)^2 - \left(\frac{dw_{0j}}{dx_j} \right)^2 \right] dx_j \quad (\text{R4})$$

and S represents the resulting axial force

$$S = \frac{EA}{L}\Delta = \frac{EA}{2L} \int_0^L \left[\left(\frac{\partial w_j}{\partial x_j} \right)^2 - \left(\frac{dw_{0j}}{dx_j} \right)^2 \right] dx_j \quad (\text{R5})$$

Substitution of Eqs. (R4) and (R5) into Eq. (R3) yields

$$V_s = \frac{1}{2}S\Delta = \frac{EA}{8L} \left[\int_0^L \left(\frac{\partial w_j}{\partial x_j} \right)^2 - \left(\frac{dw_{0j}}{dx_j} \right)^2 dx_j \right]^2, \quad (\text{R6})$$

which is identical to the last term in Eq. (R2).

Comment 1.10

I believe that in Supplemental Information, after Eq. S8: "until" should be "while".

We thank the Reviewer for this comment. We have changed the text accordingly.

Comment 1.11

Fig. S9 is difficult to understand. I am not sure what I am looking at.

We appreciate this comment and we apologize for not being as clear as desired. In Fig. S9 we are reporting the midpoint positions of each arch (*i.e.* $w_j(L/2, t)$) in a chain comprising 50 identical elements with zero damping, $\beta = 0$. For this unrealistic scenario, the signal propagates through the entire array triggering a cascade of snapping events. To facilitate the reader, we have now added to Fig. S9 a schematic of the chain (for completeness, the updated figure is also reported below as Fig. R3).

Symmetric elements – Symmetric array | No damping, $\beta = 0$

Figure R3: **Symmetric elements – Symmetric array: response of an ideal chain with no damping.** Numerical simulation of an array comprising $N = 50$ elastically deformed shallow arches all with rise $e_j = 12.4$ mm in the absence of damping (*i.e.* $\beta = 0$). An ideal system with all identical elastic elements and no damping can potentially sustain a transition wave over arbitrary distances.

Comment 1.12

Fig. S10: which lines are on the left/right side of the chain? Are the lines offset?

We thank the Reviewer for this comment. Actually, we made a mistake on the scale in one of the four plots causing a line offset. We have corrected the scale and inserted the new figure in the Supplementary Information. For completeness, we are also reporting the updated figure here as Fig. R4.

Symmetric elements – Graded and asymmetric array $|e_j| > |e_{j+1}|$

Figure R4: **Symmetric elements – Graded and asymmetric array.** Results for the same structure considered in Fig. 3 of the main text. **a**, Schematic of the structure. **b**, Comparison between the experimentally measured (thick-dotted lines) and numerically predicted (thin lines) positions of the midpoints of the arches when the system is excited left-to-right and right-to-left as indicated in (a). **c**, Schematic of the structure. **d**, Comparison between the experimentally measured (thick-dotted lines) and numerically predicted (thin lines) positions of the arches when the system is excited left-to-right and right-to-left as indicated in (c).

Response to Reviewer 2

The paper “Programming nonreciprocity and reversibility in multistable mechanical Metamaterials” by Librandi et al. reports the non-reciprocal and reversible transmission of nonlinear mechanical waves in mechanical systems consisting of bistable arches with spatially graded (a) symmetric energy barriers. In particular, the authors demonstrate that such mechanical systems can transmit nonlinear waves unidirectionally and reversibly using spatial grading. The authors further investigate the dynamics of these systems, they notably demonstrate that the nonlinear waves tend to accelerate. They also explore the design space by investigating how asymmetric bistable potentials can be used as energy sources to overcome dissipation. The paper is overall nicely written and well structured and these results will be of interests to scientists interested in nonlinear waves in mechanical systems. However, although the specific platform used by the authors is new, they are not the first to investigate spatially graded structures or asymmetric potentials and unfortunately, I don’t see where the conceptual novelty is. For these reasons, I don’t think that the paper is a good fit for Nat. Comm. and I would recommend a more specialized publication venue.

We thank Reviewer for her/his feedback. We hope he/she finds the revised version of our manuscript suitable for publication.

Comment 2.1

Transition waves in spatially graded bistable architectures with non-reciprocal wave propagation have been discussed in earlier works, see e.g. Hwang and Arrieta, PRE 2018. As acknowledge by the authors, transition waves with asymmetric potentials have been already discussed in refs [19-29,31]. In comparison to these works, I unfortunately don’t see much conceptual novelty in the present work.

We agree with the Reviewer that the propagation of transition waves in chains of couple bistable elements with asymmetric energy profile has been recently investigated in several papers [2, 4, 3, 5] including the one suggested by the Reviewer [6], which is cited in the revised version of the manuscript. Differently from previous studies that have considered arrays of purely asymmetric elements, here we investigate the dynamic response of systems comprising a mixture of symmetric and asymmetric elements. This is possible because the plastically and elastically deformed arches, despite their different energy landscape, share the same shape and, therefore, are geometrically compatible and can be easily combined to form arrays. Our platform allows us to precisely design the energy landscapes and to break the symmetry of the system both at the building block level and at the structural level, providing new opportunities for controlling nonreciprocity in nonlinear media. This important point is now emphasized on page 4 of the main text

“While previous studies have considered arrays of purely asymmetric elements [19, 31–33], here we investigate the dynamic response of systems comprising a mixture of symmetric and asymmetric elements. This is possible because the plastically and elastically deformed arches, despite their different energy landscape, share the same shape and, therefore, are geometrically compatible and can be easily combined to form arrays.”

Further, we have also extended the model initially proposed by Nadkarni *et al.* [7] to estimate the global wave speed, c_{global} , for systems made of building blocks with both symmetric and asymmetric on-site energy potentials (please see our response to Comment 3C for details).

Comment 2.2

I think that a homogenized description that would describe the continuum limit governing equation for $\psi_1(j, t)$ could be useful to predict the dynamics of transition wave and rationalize the various findings discussed in the paper. I feel that a description at such level is now lacking.

We thank the Reviewer for raising this interesting point. We fully agree that taking the continuum limit of the governing equations could be useful to predict the dynamics of transition wave and rationalize the findings discussed in the paper. However, this step is not trivial and present several challenges. For example, our building blocks snap up-to-down or down-to-up while the wave propagation occurs left-to-right or right-to-left, so that the dynamic of the building block is orthogonal with respect the direction of propagation of the signal. This makes the problem harder and unique in its kind. As such, we believe it deserves a future separate study.

Comment 2.3

I find the figures like Figs. 3d, 4c hard to parse. Contour plots would perhaps be easier to read? Also, I find Fig. 5b hard to parse with symbols having same colors in different curves.

We thank the Reviewer for this comment. Contour plots for Figs. 3d and 4c have been added to the revised Supplementary Information as Fig. S19 (for completeness, they are also reported below as Fig. R5). Further, the captions for Figs. 3d and 4c have been amended by referring to the contours plots of Fig. S19. Finally, Fig. 5b has now new colors and a more detailed legenda (its new version is also reported here as Fig. R7). We believe that this revised figure should convey messages more clearly.

Figure R5: **Comparison between experimental and numerical transition waves.** Numerically predicted position across each arch, $w_j(x_j, t)$, during the propagation of the wave in (a) the graded chain comprising elastically deformed arches tested in Fig. 3d in the left-to-right direction; (b) same graded chain tested in Fig. 3d in the right-to-left direction; (c) the chain comprising elastically and plastically deformed arches tested in Fig. 4c in the left-to-right direction (pattern showed in the schematic); (d) same chain tested in Fig. 4c in the right-to-left direction. The red dots represent the experimentally measured snapping times for each arch.

Response to Reviewer 3

Reciprocity is a generally accepted concept in classic elastic systems. However, recent studies on bistable and multistable mechanical metamaterials have shown that nonreciprocity can be realized by careful design of the microstructure constituting the metamaterials. This study demonstrates via 1D arrays of bistable arches that nonreciprocity and reversibility in the elastic wave propagation can be independently tuned. The idea is to combine the symmetry in the energy potential at element level and the geometry symmetry at structural level. Although the stable propagation of signals is a direct extension of the authors' PNAS paper (Ref. 19), the demonstrated reversible and nonreversible nonreciprocal behaviors via elastically and plastically deformed arches are interesting. The paper is thus recommended for publication in Nature Communications.

We thank the Reviewer for carefully reading our manuscript and for his/her insightful comments. We are also glad to see that he/she recommends it for publication in Nature Communications.

Comment 3.1

*In Figs. 2-4, the authors showed excellent agreement between experiment and modeling, with a particular choice of the viscous damping factor being $1.4 * 10^{-6}$ kg/(ms). It is not clear how the authors adopted such a value. Is it simply by the trial and error method?*

As suggested by the Reviewer, β was obtained by trial and error based on experimental evidence. Note that the damping parameter β represents an effective dissipation parameter which takes into account both viscous effects and dissipation at the hinges. Please note that in the previous version there was a typo in the units of β that has been fixed.

Comment 3.2

It was shown in Fig. 3e that stable propagation of signals is obtainable when the rise Δe_j is great than $150 \mu\text{m}$. The underlying mechanism is that stable propagation is possible when the dissipated energy of each unit can be fully compensated by the energy difference between adjacent units. It would be interesting if the authors can quantify relationship between the critical value of Δe_j and the viscous damping factor.

This is a good point. Following the Reviewer's suggestion, we have performed an additional numerical study where we have determined for $\beta \in [0, 1.6]$ kg/m · s the smallest Δe_j , Δe_{cr} , for which a stable propagation is possible in a chain with $N = 31$ arches. The results reported in Fig. R6 indicate that Δe_{cr} linearly increases as a function of β (i.e. $\Delta e_{cr} \sim 249\beta$). Note that Fig. R6 has been added to the revised version of SI as Fig. S20. Please note that in the previous version there was a typo in the units of β that has been fixed.

Figure R6: **Graded array and damping.** In a graded and asymmetric array the smallest difference in the rise of two consecutive arches, Δe_{cr} , does linearly depend on the viscous damping β (*i.e.* $\Delta e_{cr} \sim 249\beta$).

Comment 3.3

The authors claimed that “control on the speed of the pulses can be achieved by varying the density of plastically deformed elements in the chain”. This is true by considering the potential energy shown in Fig. 4a for a plastically deformed arch. The energy difference ΔV provides the driving source for the stable propagation. If the authors correlate ΔV with the dissipated energy in the system, they should be able to derive an analytical model for the speed of pulses.

This is another very good point. For a chain comprising only elements with asymmetric on-site energy potential the wave speed, c_{global} , can be estimated by balancing the total transported kinetic energy E_d , the difference ΔV between the higher and lower energy well, and the energy dissipated as [7]

$$c_{global} = \frac{2\beta L E_d}{\Delta V} \quad (\text{R7})$$

where β represents the viscous damping and L is the span of the arches. Further, the total transported kinetic energy can be expressed as

$$E_d = \sum_{j=1}^N \frac{1}{2} v_j^2 L, \quad (\text{R8})$$

where v_j is the speed at which the j -th arch snaps (*i.e.* its snapping-speed) and can be computed as

$$v_j = \frac{|w_j(L/2, t_{in})| + |w_j(L/2, t_{end})|}{t_{end} - t_{in}} \quad (\text{R9})$$

t_{in} and t_{end} denoting the instants of time at which the snapping of an arch starts and ends, respectively (see Fig. S15). Eq. (R7) predicts $c_{global} = 503$ units/s for a chain comprising $N = 49$ plastically deformed arches with $e_j=12.4$ mm - a prediction that closely match our numerical results (from our simulations we compute $c_{global} \sim 497$ units/s).

Following the Reviewer's suggestion, we have extended the analytical approach proposed by Nadkarni *et al.* (2016) to chains comprising both elastically and plastically deformed arches (*i.e.* building blocks that have both symmetric and asymmetric on-site energy potentials). Towards this end, we focus on a super-cell made of n arches that captures the periodic pattern of elastic/plastic arches for which E_d can be estimated as

$$E_d = \sum_{j=1}^{n_{sc}} \frac{1}{2} \left(\frac{1}{n} \sum_{i=1}^n v_{(j-1)n+i} \right)^2 nL, \quad (\text{R10})$$

with n_{sc} being the number of the super-cells in the chain. Also in this case v_j is computed using Eq. (R9) and then c_{global} is obtained by substituting Eq. (R10) into Eq. (R7). In Fig. R7 we compare the predictions of Eq. (R7) with the numerical results for chains with varying density of plastically deformed elements. We find that Eq. (R7) nicely captures the wave speed observed in our numerical simulations.

The main text has been modified to include these important findings:

“Note that c_{global} can be evaluated by balancing the total transported kinetic energy, E_d , the difference ΔV between the higher and lower energy well for the asymmetric elements, and the energy dissipated as [46]

$$c_{global} = \frac{2\beta LE_d}{\Delta V}. \quad (\text{R11})$$

where E_d can be estimated as

$$E_d = \sum_{j=1}^{n_{sc}} \frac{1}{2} \left(\frac{1}{n} \sum_{i=1}^n v_{(j-1)n+i} \right)^2 nL, \quad (\text{R12})$$

n denoting the number of arches in the super-cell that captures the periodic pattern of elastic/plastic arches and n_{sc} being the number of super-cells in the chain. Moreover, v_j is the the speed at which the j -th arch snaps (*i.e.* the snapping-speed), which can be computed as

$$v_j = \frac{|w_j(L/2, t_{in})| + |w_j(L/2, t_{end})|}{t_{end} - t_{in}} \quad (\text{R13})$$

t_{in} and t_{end} denoting the instants of time at which the snapping of the j -th arch starts and ends, respectively (see Fig. S15). In Fig. 5b we compare the predictions from Eq. R11 with the numerical results for chains with varying density of plastically deformed elements and find that the model nicely captures the wave speed observed in the simulations.”

Figure R7: **Speed per unit for different arch patterns with corresponding analytical prediction.** Local speed of the transition waves along the chain for different patterns of elastically/plastically deformed arches. The dashed lines correspond to the predictions of Eqs. (5)-(7). (i) Chain with all plastic arches, (ii) Chain with one plastic and one elastic arches, (iii) Chain with one plastic and two elastic arches, (iv) Chain with one plastic and three elastic arches, (v) Chain with one plastic and four elastic arches.

References

- [1] Sicong Shan, Sung H Kang, Jordan R Raney, Pai Wang, Lichen Fang, Francisco Candido, Jennifer A Lewis, and Katia Bertoldi. *Multistable architected materials for trapping elastic strain energy*. *Advanced Materials*, 27(29):4296–4301, 2015.
- [2] Jordan R. Raney, Neel Nadkarni, Chiara Daraio, Dennis M. Kochmann, Jennifer A. Lewis, and Katia Bertoldi. *Stable propagation of mechanical signals in soft media using stored elastic energy*. *Proceedings of the National Academy of Sciences*, 113(35):9722–9727, 2016.
- [3] Neel Nadkarni, Andres F Arrieta, Christopher Chong, Dennis M Kochmann, and Chiara Daraio. *Unidirectional transition waves in bistable lattices*. *Physical review letters*, 116(24):244501, 2016.
- [4] Lishuai Jin, Romik Khajehtourian, Jochen Mueller, Ahmad Rafsanjani, Vincent Tournat, Katia Bertoldi, and Dennis M. Kochmann. *Guided transition waves in multistable mechanical metamaterials*. *Proceedings of the National Academy of Sciences of the United States of America*, 2020.
- [5] Neel Nadkarni, Chiara Daraio, and Dennis M Kochmann. *Dynamics of periodic mechanical structures containing bistable elastic elements: From elastic to solitary wave propagation*. *Physical Review E*, 90(2):023204, 2014.
- [6] Myungwon Hwang and Andres F Arrieta. *Solitary waves in bistable lattices with stiffness grading: augmenting propagation control*. *Physical Review E*, 98(4):042205, 2018.
- [7] Neel Nadkarni, Chiara Daraio, Rohan Abeyaratne, and Dennis M. Kochmann. *Universal energy transport law for dissipative and diffusive phase transitions*. *Phys. Rev. B*, 93:104109, Mar 2016.

REVIEWERS' COMMENTS

Reviewer #1 (Remarks to the Author):

The authors have addressed well my comments. The added discussion of the results and Eq. 5, add some form of generality to the results of this manuscript. The ability of independently program reversibility and reciprocity is interesting and novel. I recommend publication in Nature Communications.

Reviewer #3 (Remarks to the Author):

The quality of the paper has been vastly improved. All my concerns have been addressed. I am happy with the changes made by the authors. The paper is recommended for publication in Nat Commun.